# The Relative Importance of Spatial Scale Variables for Explaning Macroinvertebrate Richness in Different Aquatic Ecological Function Regions

**Yuan Zhang [1,2], Xiaobo Jia [1,2], Jianing Lin [1,2], Chang Qian [1,2], Xin Gao [1,2,* and Sen Ding [1,2,*]**

[1] State Key Laboratory of Environmental Criteria and Risk Assessment, Chinese Research Academy of Environmental Sciences, Beijing 100012, China

[2] Laboratory of Riverine Ecological Conservation and Technology, Chinese Research Academy of Environmental Sciences, Beijing 100012, China

* Correspondence: gaoxin@craes.org.cn (X.G.); bearnaise@163.com (S.D.)

**Abstract:** Identifying the key drivers of aquatic fauna structuring at multiple spatial scales is critical in reducing biodiversity loss. Macroinvertebrates are the most sensitive indicators of disturbance and they are used as a cost-effective tool for bioassessment at catchment and site scales. The focus of our study was to identify the key drivers from three classes of environmental variables (geophysical landscape, land use, and site habitat) that influence macroinvertebrate richness in different aquatic ecological function regions (AEFRs) of the Liaohe River Basin. We sampled macroinvertebrate assemblages, extracted geophysical and climate variables from geospatial data, and quantified physical and chemical habitats from 407 randomly distributed sites that belong to the three level-I AEFRs. We analyzed our data through multiple linear regression models by using the three classes of predicted variables alone and in combination. The models that were constructed in the first level-I AEFR explained similar amounts of macroinvertebrate richness and had the maximum ability to explain how macroinvertebrate richness distributed (denoted "explaining ability"; geophysical landscape: $R_{GL}^2 \approx 60\%$, land use and site habitat: $R_{LU}^2$ and $R_{SH}^2 \approx 50\%$, and combined: $R_{CB}^2 \approx 75\%$). The explaining abilities for the third level-I AEFR were as follows: $R_{GL}^2 \approx 11\%$, $R_{LU}^2 \approx 14\%$, $R_{SH}^2 \approx 25\%$, and $R_{CB}^2 \approx 38\%$. The explaining abilities for the 4th level-I AEFR were as follows: $R_{GL}^2 \approx 30\%$, $R_{LU}^2 \approx 7\%$, $R_{SH}^2 \approx 40\%$, and $R_{CB}^2 \approx 55\%$. We conclude that: (1) all of the combined models explained more interaction as compared with the single models; (2) the environmental variables differed among different level-I AEFRs; and, (3) variables in the site habitat scale were the most robust explainers when analyzing the relationship between environmental variables and macroinvertebrate richness and they can be recommended as the optimal candidate explainer. These results may provide cost-effective tools for distinguishing and identifying the drivers of sensitive aquatic organisms at regional scales.

**Keywords:** geophysical landscape; land use; habitat quality; aquatic ecological function region (AEFR); anthropogenic disturbances; biodiversity; macroinvertebrate; richness; multiple linear regression model; variance partitioning

## 1. Introduction

Freshwater ecosystems provide irreplaceable ecological functions for human beings, including diluting contaminants and providing energy and water supply for irrigation and aquaculture. In addition, they maintain high biodiversity, supporting nearly 12% of the total species richness [1,2]. Nevertheless, the biodiversity of freshwater ecosystems is the most threatened, and some researchers indicate that the species extinction speed is five times higher than in every other community [3].

Species richness has been used as a useful indicator for measuring biodiversity, and identifying the distribution pattern and the key environmental variables that affect it at various spatial scales is essential in preventing biodiversity loss and formulating effective conservation actions in freshwater ecosystems [4–7]. Benthic macroinvertebrates are often used for the biological assessment of changes in response to environmental conditions—such as mobility, morphology, physiology, taxa sensitivity to disturbance, and the continuum response to environmental variables [8–13]—in freshwater ecosystems via single and multiple index and predicted models [12,14].

Researchers have claimed that the climate, geology, and topography at large scales can influence geomorphic processes and control the nutrient input, riparian habitat, and water quality at smaller scales, especially the reach and site scales, and the aquatic fauna via various and complicated pathways [15–18]. Land use, from undisturbed to human-dominated landscapes, represents anthropogenic disturbance at the catchment scale, and it has been used for decades to illustrate the direct and indirect anthropogenic effects on freshwater ecosystems [19–24]. Previous studies have documented that land use can affect the function and processes of freshwater ecosystems through various pathways, including hydrological alteration [25], riparian clearing [26], loss of woody debris [27], input of excess sedimentation, and nutrient and containment [28–30], all of which affect the quality and availability of site habitats for aquatic organisms [7]. Physical and chemical variables have a key impact on aquatic assemblages, owing to the greater direct effects of site-scale variables compared with catchment-scale [31]. However, because smaller scale variables are always structured by the larger scale variables, it is difficult to identify the relative importance of each variable on aquatic communities [7]. Nevertheless, researchers can determine the key factors that structure assemblages with the help of the increasing the availability of digitization of landscape variables at the catchment scale [7].

Previous studies have focused on the interaction of variables at various scales with aquatic organisms, especially fish and macroinvertebrates, in headwaters and the whole catchment [7,24,32]. The differences between geophysical variables, land use, and human impacts among different ecoregions have not been received much attentions in previous research [33]. Ecoregions are the management units that divide the whole catchment into different units that are based on the characteristics of territorial environmental driving factors, including landform, agrotype, climate, natural vegetation, and land use [34–39]. The ecoregions highlight the significant heterogeneity features among different aquatic ecological regions; however, similarities in water quality and aquatic organisms exist within one ecoregion. Thus, we predicted that different sets of environmental variables would explain the richness in each ecoregion.

The Liaohe River Basin, which is the seventh largest watershed in northeast China, covers more than $2.2 \times 10^5$ km$^2$, and it mainly experiences a temperate semi-humid and semi-arid monsoon climate. This watershed has become a hotspot for biodiversity conservation efforts in recent years, owing to the extinction of large endemic fish species (approximately half of the historically present species currently exist) and the loss of biodiversity that are associated with degraded water quality and physical habitat quality as a result of intense deforestation and a transformation from underdeveloped to agricultural and urban land use [40–43]. Researchers have partitioned the whole Liaohe River Basin into four level-I aquatic ecological function regions to manage the natural resources more scientifically and reasonably (hereafter, AEFRs, Figure 1, unpublished data), which are based on identifying the heterogeneous features of structure and function of aquatic ecosystems at the catchment scale. The objectives of the current paper were to identify the relative importance of environmental variables at the catchment and site scales individually and combined for macroinvertebrate richness in every AEFR of the Liaohe River Basin, and to identify the model that would best explain more variability.

## 2. Methods

### 2.1. Study Area and Site Selection

The Liaohe River Basin (40°30′–45°10′ N, 117°00′–125°30′ E) is composed of two independent separate hydrographic nets, the Da-Liao and Hun-Tai River systems, which flow into the Bohai Sea. The topography is composed of 48.2% mountains, 21.5% hills, 24.3% plains, and 6% sand, and the major stems and tributaries total 1430 km. The maximal drop in altitude is more than 1200 m, decreasing from piedmont in a west and southeast direction to the central plain, which is called the Liaohe River Plain, where the large and medium-size cities and major grain-producing areas in Liaoning Province are distributed. Annual rainfall decreases from the southeast to the northwest and ranges from 900 to 350 mm [42]. The area belonging to the second level-I AEFR has been experiencing drought for years and it is not included in the study. In the upper area of the Da-Liao and Hun-Tai Rivers, grass and forests dominate the land use, respectively, whereas the land use where the Liaohe River Plain is located is dominated by intensive agriculture and urban use [41,43].

A total of 407 sampling sites were randomly selected and distributed evenly as much as possible (Figure 1): 36, 169, and 202 in the first, third, and fourth level-I AEFRs, respectively. Macroinvertebrate, substrate, and water samples were collected in a 100 m stream reach for each site. Sampling for the first and third level-I AEFRs was conducted from late August to late September in 2012 and 2013, and for the fourth level-I AEFR sampling was conducted from May 2009 to August 2010.

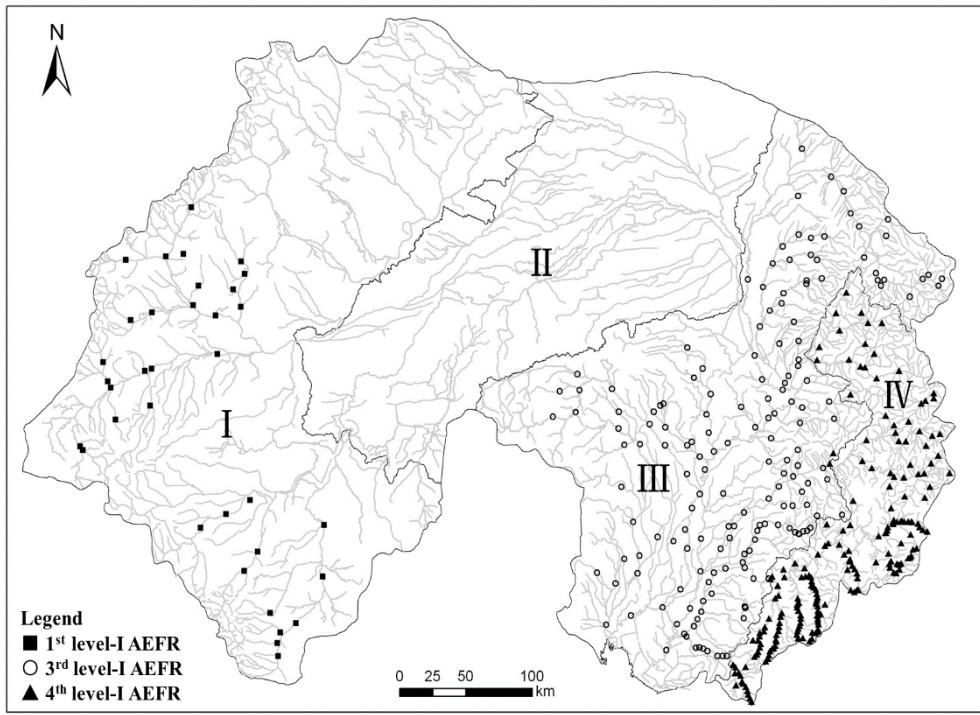

**Figure 1.** Map of study area and distribution of 407 sampling sites.

### 2.2. Geophysical Landscape Variables

The accurate point coordinates and altitude for each site were measured using a portable GPS (Explorist-200, Magellan, San Dimas, CA, USA) in situ. The slope, sinuosity, and stream order were calculated from the digital elevation model (DEM) datum within 90 m resolution and a geomorphic type vector diagram of Liaohe River Basin at a scale of 1:100,000. The DEM datum was downloaded from the USGS website (http://srtm.usgs.gov). The drainage network of the Liaohe River Basin was extracted from a DEM based on a ground water overflow model. All of the rivers were divided into segments on the basis of the setting of an intersection as the break point, and the intersection was determined by

visualization. Reach length, slope, sinuosity, and stream order for each site were obtained from the specified river segment. The slope was automatically extracted by inputting relevant data into the slope calculation tool in surface analysis based on the DEM layers data. Sinuosity was calculated by the ratio value between the physical and straight length for each river segment. The Strahler river classification method calculated stream order. The average rainfall and average temperature were extracted from the climate data at the national scale, which were collected from 1657 climate monitoring stations. Each station has annual rainfall and temperature data lasting ≥20 years. The rainfall and temperature data of the studied areas were interpolated while using the inverse distance weighted method. The grid cell value of mean rainfall and temperature were transferred to each site.

## 2.3. Land Use

The land use features were extracted from Landsat-5 remote sensing images. Data were gathered before September 2007 to reduce the impact of crop harvest in surrounding farmland [41]. The catchment landscape was classified into nine categories, including forestland, grassland, farmland, constructed land, river, reservoir, bottomland, wetland, and fishpond after the pretreatment of the images within refined geometric rectification, image fusion, capture using ENVI 4.4 (Exelis Visual Information Solutions, Inc., CO, USA), and importation into Definiens Developer 7.0 (Definiens AG, Munich, Germany). The final interpreted results were determined by amending the preliminary scheme on the basis of a combination of field investigation and geomorphology characteristics by using a man-machine interaction method between artificial interpretation and the closest classification method. Thus, the preliminary nine land use classifications were revised into six categories, which included unused land, water area, forestland, constructed land, farmland, and grassland.

## 2.4. Site Habitat Variables

Before sampling macroinvertebrates, 15 physical and chemical variables in site-scale (Table 1) were measured to describe the habitat quality within a 100 m reach at each site in situ. Two physical variables were measured by field measurement (mean depth, m) and portable instrument (velocity, m/s, FP101, Geotech Environmental Equipment, Inc., Denver, CO, USA). Six water chemical variables were measured by portable instrument (YSI-80, YSI Inc., Yellow Springs, OH, USA), which included water temperature (°C), electric conductivity (µS/cm), total dissolved solids (mg/L), pH, and dissolved oxygen (mg/L), and three variables were sampled following standard methods (Environmental Quality Standards for Surface Water, GB 3838-2002) (SEPA and AQSIQ, 2002), including total nitrogen (TN, mg/L), total phosphorus (TP, mg/L), and ammonia nitrogen ($NH_3$–N, mg/L). Based on the criteria of Jiang et al. (2010) [44], dominant substrates were classified and assigned into one of five types: (1) Boulder, (2) Cobble, (3) Pebble, (4) Gravel, and (5) Sand/Silt.

## 2.5. Macroinvertebrate Sampling

A total of 1628 macroinvertebrate samples, one qualitative and three quantitative samples for each site, were collected. All of the qualitative samples were collected while using a D-net (with 500 µm mesh size) across as many habitats as possible with different velocities, water depths, and substrates. The quantitative samples were collected while using a Surber net (0.09 $m^2$ in area, with 500 µm mesh size) for wadeable streams, and a bottom sampler with fixed opening area (1/16 $m^2$) for non-wadeable rivers. This combined sampling method has been proven to represent macroinvertebrate taxonomic richness sufficiently, precisely and accurately enough at both the catchment and site scale [7]. All of the specimens were picked in situ and preserved with 10% buffered formalin solution. In the laboratory, the specimens were identified to the lowest feasible taxonomic level.

## 2.6. Data Analysis

Multiple linear regression (MLR) models were used to structure the interactive correlation between the environmental variables, which were in different scales and types, and taxonomic richness [7,24,45].

MLR models were conducted by SPSS statistical software (version 16.0, SPSS Inc., IL, USA). Based on the criteria of Ferreira et al. (2014) [24] and Macedo et al. (2014) [7], all 32 environmental variables were classified into two scales and three levels (Table 1). Environmental variables belonged to geophysical landscape level represented the physical position for each site which were not controlled by human activities, and the variables belonged to land use and site habitat levels all represented the status could changed by anthropogenic activities.

**Table 1.** Three levels of environmental variables used to establish the multiple linear regression (MLR) models in two scales.

| Scale | Level | Variable | Method | Code |
|---|---|---|---|---|
| Catchment | Geophysical landscape | Longitude | *in situ* | Longitude |
| | | Latitude | *in situ* | Latitude |
| | | Altitude | *in situ* | Altitude |
| | | Slope | GIS | Slope |
| | | Sinuosity | GIS | Sinuosity |
| | | Stream Order | GIS | Streamorder |
| | | Reach Length | GIS | Length |
| | | Distance from Mouth | GIS | DisfrMouth |
| | | Average Temperature | GIS | AverTem |
| | | Average Rainfall | GIS | AverRain |
| | | Up-catchment Area | GIS | UpcatchArea |
| | Land use | Un-used Area | GIS | UnusedArea |
| | | Water Area | GIS | WaterArea |
| | | Forest Area | GIS | ForestArea |
| | | Constructed Area | GIS | ConstrArea |
| | | Crop Area | GIS | CropArea |
| | | Grass Area | GIS | GrassArea |
| Site | Site habitat | Depth | *in situ* | Depth |
| | | Velocity | *in situ* | Velocity |
| | | Boulder | *in situ* | Boulder |
| | | Cobble | *in situ* | Cobble |
| | | Pebble | *in situ* | Pebble |
| | | Gravel | *in situ* | Gravel |
| | | Sand/Silt | *in situ* | Sand/Silt |
| | | Water Temperature | *in situ* | WaterTem |
| | | Electric conductivity | *in situ* | EC |
| | | Total Dissolved Solids | *in situ* | TDS |
| | | pH | *in situ* | pH |
| | | Dissolved Oxygen | *in situ* | DO |
| | | Total Nitrogen | in situ & lab | TN |
| | | Total Phosphorus | in situ & lab | TP |
| | | Ammonia Nitrogen | in situ & lab | $NH_3$-N |

Model construction followed the criteria that were cited by Macedo et al. (2014) [7]. First, to meet normality, the proportional variables were arcsine squared root transformed and the continuous variables were log transformed. Second, the transformed variables were screened by the following standards: (1) variables with >90% zero values were eliminated; (2) Pearson correlation was conducted to identify the highly correlated variables ($|r| > 0.8$), in order to preserve the variables with more ecological meaning. For example, longitude was highly positively correlated with DisfrMouth ($r = 0.916$) in model construction for the fourth level-I AEFR. The two variables all described the spatial relative location and we consider DisfrMouth contains more comprehensive meaning, not only the physical location, but also the relative habitat quality for the sampling site. DisfrMouth was always be an important environmental driver in other relevant researches [46–48]; and, (3) the candidate variables were rejected if they had a low correlation coefficient ($|r| < 0.1$) between the remaining variables and macroinvertebrate richness by Pearson correlation analysis. Third, three separated MLR models were established for the geophysical landscape variables, land use variables, and site habitat variables

singly; thus, the optimal explained level could be identified through the regression analyses (forward selection, *P*-to enter = 0.15). The Durbin-Watson values (D-W values) and histograms of standardized residuals were calculated and charted to test the validity for each model. Fourth, a combined model for all level environmental variables using partial linear regression was established and used to evaluate the relative importance of each environmental level on the richness [7,49].

## 3. Results

*3.1. Characteristics of Assemblage Richness and Environment Variables among the Three Level-I AEFRs*

274 macroinvertebrate taxa were identified for the whole Liaohe River Basin, with the taxa number occurring in the first (66 taxa), third (151), and fourth (218 taxa) level-I AEFRs differing considerably (see Table S1). Variables were significantly different among all of the AEFRs, except Length and NH$_3$-N (Table 2). The predominant geographical landscape characteristics in the 1st level-I AEFR were high altitude, slope, distance from mouth, and lowest average rainfall. The unused area was the lowest proportion in the first level-I AEFR and the predominant land use was grassland. Although constructed and cropland use were not dominant land use types, owing to the severe soil erosion in Inner Mongolia, the substrate was characterized by a large proportion of sand, analogously to the characterization in the third level-I AEFR, where cropland accounted for the greatest land use.

**Table 2.** Comparisons of assemblage richness and environmental variables (means ± SD) among different aquatic ecological function regions (AEFRs).

| Variables | AEFRs | | | *p*-Value |
|---|---|---|---|---|
| | 1st | 3rd | 4th | |
| Assemblage Richness | 10.90 ± 4.36 | 10.68 ± 5.77 | 17.32 ± 11.95 | <0.01 [a,c] |
| Altitude | 816.50 ± 228.58 | 91.84 ± 75.18 | 265.84 ± 125.60 | <0.01 [a,c] |
| Slope | 5.78 ± 4.24 | 2.19 ± 3.56 | 8.22 ± 7.85 | <0.01 [a,c] |
| Sinuosity | 1.16 ± 0.11 | 1.30 ± 0.22 | 1.32 ± 0.32 | <0.01 [a,c] |
| Streamorder | 3.00 ± 1.11 | 3.07 ± 1.48 | 2.35 ± 1.15 | <0.01 [a,b] |
| Length | 11.38 ± 10.00 | 12.84 ± 13.00 | 12.81 ± 11.26 | 0.531 [b] |
| DisfrMouth | $119.32 \times 10^4 \pm 4.91 \times 10^4$ | $33.70 \times 10^4 \pm 19.94 \times 10^4$ | $37.00 \times 10^4 \pm 9.79 \times 10^4$ | <0.01 [a,c] |
| AverTem | 4.10 ± 2.08 | 7.64 ± 1.11 | 5.78 ± 1.52 | <0.01 [a,c] |
| AverRain | 443.09 ± 71.95 | 645.63 ± 72.96 | 867.74 ± 73.85 | <0.01 [b] |
| UpcatchArea | $20.24 \times 10^8 \pm 25.26 \times 10^8$ | $147.03 \times 10^8 \pm 450.11 \times 10^8$ | $5.92 \times 10^8 \pm 10.88 \times 10^8$ | <0.01 [a,c] |
| UnusedArea | 8.99 ± 11.28 | 1.11 ± 6.02 | 0.16 ± 0.87 | <0.01 [a,c] |
| WaterArea | 4.04 ± 4.26 | 8.34 ± 12.64 | 2.05 ± 4.00 | <0.01 [a,c] |
| ForestArea | 7.78 ± 10.68 | 11.22 ± 16.96 | 66.98 ± 16.99 | <0.01 [a,c] |
| ConstrArea | 4.47 ± 5.02 | 10.33 ± 8.50 | 2.67 ± 3.58 | <0.01 [a,c] |
| CropArea | 31.16 ± 25.83 | 67.22 ± 19.42 | 26.93 ± 13.75 | <0.01 [a,c] |
| GrassArea | 43.56 ± 25.59 | 1.78 ± 4.60 | 1.21 ± 2.17 | <0.01 [a,c] |
| Depth | 19.69 ± 13.74 | 30.78 ± 17.14 | 22.56 ± 10.15 | <0.01 [a,c] |
| Velocity | 1.51 ± 0.96 | 0.54 ± 0.62 | 0.51 ± 0.39 | <0.01 [a,c] |
| Boulder | 20.26 ± 24.71 | 9.99 ± 19.81 | 39.44 ± 23.41 | <0.01 [a,c] |
| Cobble | 12.61 ± 15.83 | 7.14 ± 12.10 | 19.32 ± 10.86 | <0.01 [a,c] |
| Pebble | 15.12 ± 17.21 | 10.52 ± 15.64 | 23.26 ± 13.31 | <0.01 [a,c] |
| Gravel | 6.52 ± 10.93 | 5.74 ± 9.43 | 9.66 ± 8.25 | <0.01 [b] |
| Sand/Silt | 45.49 ± 47.27 | 62.46 ± 42.79 | 8.33 ± 14.77 | <0.01 [a,c] |
| WaterTem | 20.58 ± 4.56 | 20.44 ± 3.42 | 17.11 ± 4.73 | <0.01 [a,c] |
| EC | 373.54 ± 109.02 | 494.61 ± 181.18 | 233.25 ± 154.66 | <0.01 [b] |
| TDS | 180.14 ± 53.23 | 309.62 ± 129.27 | 161.17 ± 106.19 | <0.01 [a,c] |
| Ph | 8.35 ± 0.37 | 8.13 ± 0.52 | 8.46 ± 0.59 | <0.01 [b] |
| DO | 8.02 ± 1.56 | 8.53 ± 2.42 | 11.19 ± 3.85 | <0.01 [a,b] |
| TN | 2.62 ± 1.83 | 3.96 ± 5.18 | 3.45 ± 2.91 | <0.05 [a,c] |
| TP | 0.11 ± 0.14 | 0.24 ± 0.38 | 0.11 ± 0.18 | <0.01 [a,c] |
| NH$_3$-N | 0.67 ± 0.95 | 1.10 ± 2.56 | 0.46 ± 0.69 | 0.065 [a,c] |

[a] indicates the variable was $\log_{10}(1 + x)$ transformed, [b] indicates the one-way analysis of variance (ANOVA), [c] indicates the Kruskal-Wallis analysis.

The third level-I AEFR was characterized by low altitude, slope, and distance to mouth for the main stems of the Liaohe River flowing through the Liaohe River Plain and affluxing into the Bohai Sea. The average temperature was the highest and the rainfall stayed in the median level of the whole basin. Almost all of the modern cities are located along the main rivers of this AEFR; thus, constructed and crop land were the dominant land use type. There was severe riparian erosion that was caused by intense agriculture, and the substrate and water quality (EC, TDS, and TN) were in the worst state. Almost all of the sites of the fourth AEFR were located in the mountains, and the geological landscape and land use were characterized by high rainfall and forestland. On account of less anthropogenic disturbance, the physical and chemical variable remained optimal, with the substrate dominated by boulder and cobble, and the lowest EC and TDS and the highest DO levels present.

### 3.2. MLR Models

In the first level-I AEFR, eight variables were eliminated by the screening criteria. The MLR models established with the 24 residual variables belonging to geophysical landscape, land use, and site habitat explained different amounts of variation in macroinvertebrate richness. The variables in land use and site habitat explained similar amounts of variance in macroinvertebrate richness: ≈50% (Table 3). The geophysical landscape model explained approximately 60% variance and it was composed of DisfrMouth and UpcatchArea (negative). The land use model was composed of UnusedArea and WaterArea (all negative). The site habitat model was composed of TN (positive) and three negative variables, including TP, Depth, and Sand/Silt.

**Table 3.** Geophysical landscape, land use, and site habitat MLRs of macroinvertebrates in the first level-I AEFRs.

| AEFR | Variable | $\beta$ | Std-Error | $R^2$ | D-W Value |
|---|---|---|---|---|---|
| 1st | Geophysical landscape | | | 0.602 ** | 2.339 |
| | DisfrMouth | 5.220 | 1.204 | | |
| | UpcatchArea | −0.117 | 0.041 | | |
| | Land use | | | 0.503 ** | 1.971 |
| | UnusedArea | −0.412 | 0.094 | | |
| | WaterArea | −0.573 | 0.158 | | |
| | Site habitat | | | 0.505 ** | 1.876 |
| | TP | −1.154 | 0.525 | | |
| | TN | 0.211 | 0.112 | | |
| | Depth | −0.153 | 0.078 | | |
| | Sand/Silt | −0.056 | 0.038 | | |
| 3rd | Geophysical landscape | | | 0.112 ** | 1.588 |
| | AverRain | −1.502 | 0.431 | | |
| | UpcatchArea | −0.062 | 0.022 | | |
| | Length | −0.088 | 0.058 | | |
| | Land use | | | 0.141 ** | 1.472 |
| | WaterArea | −0.315 | 0.089 | | |
| | ConstrArea | −0.459 | 0.160 | | |
| | UnusedArea | −0.330 | 0.185 | | |
| | Site habitat | | | 0.246 ** | 1.843 |
| | $NH_3-N$ | −0.173 | 0.122 | | |
| | Boulder | 0.115 | 0.078 | | |
| | TN | −0.174 | 0.073 | | |
| | Sand/Silt | −0.088 | 0.041 | | |

**Table 3.** *Cont.*

| AEFR | Variable | $\beta$ | Std-Error | $R^2$ | D-W Value |
|------|----------|---------|-----------|-------|-----------|
| 4th | Geophysical landscape | | | 0.302 ** | 1.206 |
| | AverRain | 1.108 | 0.528 | | |
| | AverTem | −0.882 | 0.166 | | |
| | Latitude | −16.851 | 3.918 | | |
| | Length | −0.083 | 0.046 | | |
| | Land use | | | 0.065 ** | 0.847 |
| | ForestArea | 0.355 | 0.094 | | |
| | Site habitat | | | 0.395 ** | 1.517 |
| | Sand/Silt | −0.390 | 0.079 | | |
| | Velocity | −0.906 | 0.164 | | |
| | TN | −0.256 | 0.055 | | |
| | WaterTem | 0.417 | 0.115 | | |
| | TP | −0.527 | 0.266 | | |
| | Boulder | 0.111 | 0.061 | | |

** $p < 0.01$.

The 19 residual variables were used to establish the models in three levels in the third level-I AEFR. The explaining abilities of models in the third AEFR were the lowest among all of the AEFRs. The site habitat model (25%) explained the most variance, and the geophysical landscape model (11%) explained the least, whereas the land use model (14%) explained a moderate amount (Table 3). AverRain, UpcatchArea, and Length constituted the geophysical landscape model, and all of them were all negative. The land use model was composed of WaterArea, ConstrArea, and UnusedArea (all negative). Boulder had the key effect in the site habitat model, while $NH_3$–N, TN, and Sand/Silt were all negative in the model.

By the eliminated criteria, 13 variables were screened out in the fourth level-I AEFR. The land use model represented the lowest explaining ability among allof the models, only approximately 7% of the variance in richness (Table 3), whereas ForestArea (positive) was the key explainer The geophysical landscape and site habitat models explained approximately 30% and 40%, respectively. The elements in the geophysical landscape model were AverRain (positive), AverTem, Latitude, and Length. In the site habitat model, six variables were used, including WaterTem and boulder, which were positively associated with richness, and sand, velocity, TN, and TP, which were negatively associated with richness.

*3.3. Relative Importance of Variables in the Combined MLR Models*

The combined model in the first level-I AEFR explained ≈75% of the macroinvertebrate richness and was composed of DisfrMouth, TP, depth, CropArea, and AverTem (Table 4). The variance partitioning analysis indicated that geophysical landscape explained ≈12% singly, land use ≈5%, and site habitat ≈1% of the macroinvertebrate richness. Geophysical landscape and land use shared ≈37%, geophysical landscape and site habitat shared ≈41%, and land use and site habitat shared ≈39% of the explained variance. Approximately 30% of the explained variance was shared among all the three levels of variables.

The combined model in the third level-I AEFR explained ≈38% of the macroinvertebrate richness, and it was composed of three geophysical landscape variables (AverRain, Sinuosity, and DisfrMouth), three land use variables (WaterArea, ForestArea, and CropArea), and five site habitat variables ($NH_3$-N, Boulder, TN, pH, and DO) (Table 4). Variance partitioning analysis indicated that geophysical landscape explained ≈5% singly, land use ≈11%, and site habitat ≈12% of the macroinvertebrate richness. Geophysical landscape and land use shared <1%, geophysical landscape and site habitat shared ≈9%, and land use and site habitat shared ≈6% of the explained variance. Approximately 3% of the explained variance was shared among all three levels of variables.

**Table 4.** Partial linear regression for macroinvertebrate richness and all environmental variables and shared explained variances.

| AEFR | Geophysical Landscape (GL) | | | | Land Use (LU) | | | | Site Habitat (SH) | | | | Total $R^2$ | Shared Explained Variance | | | |
|---|---|---|---|---|---|---|---|---|---|---|---|---|---|---|---|---|---|
| | | $\beta$ | Std-Error | $R^2$ | | $\beta$ | Std-Error | $R^2$ | | $\beta$ | Std-Error | $R^2$ | | GL + LU $R^2$ | GL + SH $R^2$ | LU + SH $R^2$ | Combined $R^2$ |
| 1st | | | | 0.123 | | | | 0.046 | | | | 0.011 | 0.745 | 0.371 | 0.408 | 0.386 | 0.300 |
| | DisfrMouth | 4.461 | 1.151 | | CropArea | 0.170 | 0.078 | | TP | −0.925 | 0.419 | | | | | | |
| | AverTem | −0.174 | 0.105 | | | | | | Depth | −0.171 | 0.065 | | | | | | |
| 3rd | | | | 0.054 | | | | 0.111 | | | | 0.121 | 0.378 | −0.004 | 0.091 | 0.063 | 0.029 |
| | AverRain | −1.088 | 0.499 | | WaterArea | 0.009 | 0.116 | | NH$_3$-N | 0.056 | 0.138 | | | | | | |
| | Sinuosity | −0.871 | 0.510 | | ForestArea | 0.541 | 0.123 | | Boulder | 0.111 | 0.065 | | | | | | |
| | DisfrMouth | −0.119 | 0.080 | | CropArea | 0.431 | 0.149 | | TN | −0.213 | 0.089 | | | | | | |
| | | | | | | | | | pH | −2.236 | 0.912 | | | | | | |
| | | | | | | | | | DO | 0.242 | 0.147 | | | | | | |
| 4th | | | | 0.104 | | | | | | | | 0.245 | 0.547 | 0.065 | 0.150 | 0.017 | 0.017 |
| | AverRain | 0.584 | 0.463 | | | | | | Sand/Silt | −0.258 | 0.071 | | | | | | |
| | Latitude | −17.835 | 4.598 | | | | | | TN | −0.277 | 0.061 | | | | | | |
| | AverTem | −0.830 | 0.150 | | | | | | WaterTem | 0.527 | 0.113 | | | | | | |
| | Slop | 0.059 | 0.041 | | | | | | Velocity | −0.442 | 0.180 | | | | | | |
| | | | | | | | | | Boulder | 0.095 | 0.055 | | | | | | |
| | | | | | | | | | Depth | 0.169 | 0.078 | | | | | | |
| | | | | | | | | | NH$_3$-N | −0.207 | 0.111 | | | | | | |

The combined model in the fourth level-I AEFR explained ≈55% of the macroinvertebrate richness, and it was composed of four geophysical landscape variables (AverRain, Latitude, AverTem, and Slope), seven site habitat variables (Sand/Silt, TN, WaterTem, Velocity, Boulder, Depth, and $NH_3$–N), and no land use variable entered the model (Table 4). Variance partitioning analysis indicated that geophysical landscape explained ≈10% singly, site habitat ≈25%, and land use did not explain any variance of the macroinvertebrate richness. Geophysical landscape and land use shared ≈7%, geophysical landscape and site habitat shared ≈15%, and land use and site habitat shared ≈2% of the explained variance. Approximately 2% of the explained variance was shared among all three levels of variables.

## 4. Discussion

Our results showed that the key environmental variables that affect macroinvertebrate assemblages substantially differed in level-I AEFRs of the Liaohe River Basin despite the relatively weak explaining ability of some MLRs, and they were comparable to findings from other studies. We established three levels of MLR for macroinvertebrate richness at the catchment and site scales based on the geographic information and anthropogenic disturbance (changed land use) extracted from remotely sensed data and rigorous site habitat features sampled according to standard procedures. We believe that this work provides accurate and repeatable results involving multiple interactions between environmental variables and the structure of aquatic assemblages [41].

Aquatic assemblages always show differentiation among different geographic zonings, owing to the heterogeneity of site habitat and water quality, which are caused by geomorphology and climate changes at the catchment or larger scale [50–54]. The geophysical landscape MLRs in the three level-I AEFRs were different with its own features. The maximum explaining ability of the MLR alone all occurred in the first level-I AEFR where the West-Liao River and Lao-ha River and tributaries flow. The physical position for each site was the key explainer in the MLR of geophysical landscape level. DisfrMouth (positive) and UpcatchArea (negative) with richness explained the similar effect, in which the closer to the source, the better the retained site habitat quality and the more biodiversity maintained [55,56]. AverRain had a large contribution in explaining the interaction between the geophysical landscape and richness both in the third level-I (negative) and fourth level-I (positive) AEFR. Although the annual rainfall in the former AEFR was less than that in the latter, there was more possibility that a freshet might occur in the Liaohe Plain, and some other process accompanying the freshet could decrease the richness [7]. Some research has claimed that rainfall has a positive effect on aquatic organisms (fish richness) at the global scale, and at some smaller catchment scales for macroinvertebrate richness [57]. The key mechanism of the promoting effect on richness might be the dilution and flushing of contaminants to make water quality better and determining the quantity and persistence of surface water in the particular locality, especially the valley region [57,58]. Reach length also entered the two landscape models and it acted as the same negative explainer in the third and fourth level-I AEFR. In this study, sinuosity was derived by the ratio of reach length and the distance between the two points of the reach. We hypothesized that longer length might correlate with lower richness, because the elongated reach increased the relative distance between the two nodes of a reach; thus, the sinuosity decreased, especially in the mountainous region. Sinuosity can also affect the inundate process of freshets and then shape the channel patterns to generate more habitat diversity to sustain higher biodiversity. Another reason for this effect may be that longer reaches flow through a grater area, and water quality decreases as the accommodating ability of the containment increases.

Different level-I AEFRs had different explaining abilities regarding land use and macroinvertebrate richness; the strongest explaining ability was found in the first level-I AEFR. More open areas, with less shield, have similar results. In the third level-I AEFR, WaterArea, which is related to the area of reservoir up-stream, was a key variable in the MLR. Dam construction can alter the hydrological process and flow regimes, and subsequently degrade the quality and quantity of habitat and detrimentally affect the biodiversity of aquatic organisms, especially the fish species and communities [59–62]. In turn, forestland promoted biodiversity in the mountainous region, as shown in the fourth model.

Except for the model for the first level-I AEFR, the site habitat variables explained the most variance in the latter models. Previous studies have reported that physical and chemical variables at the site scale play a primary explaining role in affecting the assemblage structure and composition of lotic species [18,42–44]; this conclusion was confirmed by our results. In the three MLR models, the variables representing the microhabitat, substrate structure, and conventional physicochemical variables were all included in the explainer list. Substrate structure, especially the composition of diverse cobblestone, has been shown to be the key factor that influences macroinvertebrate structure [42]. Boulder substrate can be of crucial importance in the distribution of benthic macroinvertebrates due to the abundant refuges, which buffer populations against a variety of abiotic perturbations or biotic interactions [63]. Furthermore, boulders support some irreplaceable functions, such as providing a protective nursery for small larvae and plant detritus or periphyton trapping, which can promote the taxonomic richness of patch communities [44]. However, sand played a negative role in our models. Excess sand that is imported to rivers can result in physical damage to insect larvae and change the composition of a river bed [64]. The average size of particles becomes smaller where there are inputs of fine sediment to rivers. The stability and fluidity of the river bed may be reduced due to the interstices between larger particles become filled [65]. Most macroinvertebrate species tend to live within a specific habitat and they avoid patches that fail to meet these requirements [66–68]. Some species make a living by adhering to substrate within a relatively cleared surface; however, the abundance and richness will decrease due to the unstable surface that is caused by erosion and abrasion [69,70]. Nutrient enrichment of aquatic ecosystems, as represented by the variables of TN and TP in the current study, has been shown to have dual functions. Some results show that increased TN has a negative effect on aquatic fauna [71]; however, others have found that TN can be a promoter to some moderately tolerant species. The changed food quality and quantity due to the bottom-up controls [72–74], like the increased periphyton abundance [75], may be the reason for these discrepant findings.

The three combined models all had more explaining ability than the model based on a single variable type. In the first AEFR, the interactions among geophysical landscape, land use, and site habitat were found to be larger than the other two combined models (75% vs. 38~55%). The DisfrMouth, TP, and Depth were confirmed to be robust explainers in this region; all were included both in the single geophysical landscape and combined model. CropArea displaced the two land use types in the single land use model. The effect of agriculture on freshwater ecosystems is still far from clear. Numerous researches have claimed that the degradation of water quality and physical habitat support fewer sensitive species of fish and insect, and ultimately lead to a serious biodiversity loss [18,76–78]. These negative effects are usually caused by excess inputs of nutrients, pollutants, and pesticides, loss of riparian shading, and altered flow regimes [18,79,80]. However, with the increase of light and temperature, nutrients can also promote algal biomass, and the macroinvertebrate richness and abundance will increase, owing to the changed food web [81,82]. Variance that was explained by both the two level variables and the combined ones was close to the total explaining ability in the third and fourth level AEFR models (30% to 40%) and much higher than other relative studies (Macedo et al., 2014 [7]. The maximum value was only 10% (see Table 3 in their paper), which indicated that the synergetic effects from landscape to site scale overlapped. These results confirmed the conclusion that the processes at larger spatial scales affect the ecological features at the smaller scale [7,15]. Interestingly, the land use variables were excluded from the combined model in the fourth level AEFR; the same result was also found in the single land use model that the weakest explaining ability was for ForestArea. The relationship between land use and richness was not detected or only weakly detected, which was possibly because of covariance effects or the lower variability in Hun-Tai up-streams [7].

In this paper, we have used a range of variables at multiple levels and scales in different AEFRs to identify the key explainers in the separated models and the relative importance in structuring the macroinvertebrate richness in combined models. Our combined models did promote the explaining ability when compared to the single ones (from 7–60% to 38–75%). With the stable explaining ability of site habitat variables, we recommend that the site habitat variables, including substrate composition

and water chemical quality should be used as the robust explainers when analyzing the relationship between macroinvertebrates and environmental features.

**Supplementary Materials:** The following are available online at http://www.mdpi.com/2073-4441/11/8/1550/s1, Table S1: A total of 247 identified taxa list and the AEFR where each taxon occured in.

**Author Contributions:** Writing—original draft preparation, Y.Z. and X.G.; formal analysis, X.J. and X.G.; investigation, J.L. and C.Q.; writing—review and editing, X.G. and S.D.

**Funding:** This work was financially supported by the Major Science and Technology Program for Water Pollution Control and Treatment (No. 2012ZX07501-001).

**Acknowledgments:** The authors are greatly indebted to Xuwang Yin, Qingnan Li for their help in assaying the physicochemical parameters.

**Conflicts of Interest:** The authors declare no conflict of interest. The funders had no role in the design of the study; in the collection, analyses, or interpretation of data; in the writing of the manuscript, or in the decision to publish the results.

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
