# Peer review of "The Relative Importance of Spatial Scale Variables for Explaning Macroinvertebrate Richness in Different Aquatic Ecological Function Regions"

_water, doi:10.3390/w11081550_

Reviewer 1 Report

Please revise your paper to include tables 1 and 3, both were missing. I also encourage you to submit your paper to an English language review.

You may also wish to address how sampling two eco-regions in August-September and a third in May might have influenced your results.

Author Response

Response to Reviewer 1 Comments

Point 1: Please revise your paper to include tables 1 and 3, both were missing.

Response 1: The new revised article have 4 tables, and all of them are included.

Point-2: You may also wish to address how sampling two eco-regions in August-September and a third in May might have influenced your results.

Response 2: The purpose of the current article is to analyse the drivers to structuring the macroinvertebrate assembles in different Eco-regions and spatial scales, and seasonal difference may be analyzed with supplement of data in April and November in the future.

Point-3: I also encourage you to submit your paper to an English language review.

Response 3: The current article is already polished by an language polishing organization.

Reviewer 2 Report

The authors were invited to see the comments in the manuscript reviewed.

The manuscript requires major revisions, anywhere the euristic value of the paper is clearly unsufficient.

Collected data (geophysical, physico-chemical and biological data) must be reported at least in synthesis with figure and/or tables, in order to display the real environmental status of the investigated areas. Statistical analysis must be only a mathematical way to support the results of the collected data. The manuscript must be rewrited and improved.

Author Response

Response to Reviewer 2 Comments

Point 1: No table to display the environmental variables data, which is pointed in the Line 137 of the original article.

Response 1: The statistic comparison for all the environmental variables among different Eco-regions are showed in the Table 2.

Point 2: Have you collected 12x407=4884 samples for macroinvertebrates? Have identified all animals in these samples, which is pointed in the Line 149.

Response 2: Just one qualitative and three quantitative samples for each site have collected. And all the samples were identified.

Point 3: No list of taxa neither quantity which is pointed in the Line 186.

Response 3: All the identified taxa is listed in attached table and the macroinvertebrate assemblage richness was elected to represent the community structure. The purpose is to analyse the relationship between environmental variables and macroinvertebrate assemblage, the authors think it unnecessary to show the quantity of 274 identified taxa.

Point 4: Captions of Fig. 2 are illegible, so the text is incomprehensible, which is pointed in the Line 196.

Response 4: to show the results more clearly, figure 2 was deleted and all the analysis results are contained in Table 2 which include the comparisons for assemblage richness and environmental variables among different Eco-regions.

Point 5: Chapters 3.1 -3.2 - 3.3 are a long and tedious list of numbers, most part of them not represented in Table 2.

Response 5: Chapter 3.2 was deleted because the content is already included in the modeling process and all the analysis results are already included in the table 2 to 4, also it is English polished.

Point 6: As are lacking in the results both figures and tables on geomorphological data, physico-chemical data and macroinvertebrate data, discussion is based only on a too much long list of statistical data that are represented only in a smallest part, and for this reason appear confused. Neither are cited previous papers of the authors on basic data on which the statistical analysis is applied. So, don't appear the real differences morphological, physico-chemical and biological among the three studied areas. So the conclusions are substantially undefined and it's not clear what are the novelties emerged from the present study. Some conclusions appear to be obvious and well known concepts.

Response 6: All the lacking analysis results are already added in the new revised article which can be found in table 2 and 3 and the discussion has been revised according to the recommendations. The writing thought for discussions of the current article is stated as follows. The purpose of the article is to distinguish the key drivers which influenced the aquatic organisms according to the AEFRs. AEFRs were some useful tool to environment administrative departments which shaped by the natural and some anthropogenic (land use) variables within larger scale, the catchment scale. And the influencing mechanism for degradation of aquatic ecosystem have been stated clearly that terrestrial and aquatic changes have made a combined affection for the biodiversity lose. Management organization could pay more attentions to the key environmental variables which were elected by models, to make more efficient works for the maintenance and restoration for aquatic biodiversity. We have elected the key drivers for each AEFR, and we have stated and compared the different drivers with our finding and some other similar works in other basins around the world.

Round  2

Reviewer 1 Report

I have read the version of your paper that was resubmitted for publication;. I could find little evidence, aside from including two tables missing from the previous  paper, that you had undertaken and revisions at all.

In response to my earlier comment that you seek assistance with English expression, you responded that you had employed an English "polishing" service. I am not familiar with these services. However, it is obvious that they did not help you. Many sentences and phrases remain problematic and difficult to understand.  

Your response to my other substantive question, how extended drought in one of three sub-basins, was also less than satisfactory. I remain skeptical that this would not have in influence on variables affecting biodiversity.

Overall, I believe you did not take review comments as serious. Because the paper has not been improved from the initial form I cannot recommend publication.

Author Response

Response to Reviewer 1 Comments

The problematic sentences and phrases are revised as below.

Point-1: In Line 24-26, the original sentence “The models constructed in the 1st level-I AEFR explained similar amounts of macroinvertebrate richness and attained the maximum explaining ability (geophysical landscape: RGL2 ≈ 60%, land use and site habitat: RLU2 and RSH2 ≈ 50%, and combined: RCB2 ≈ 75%).“ revised as below.

Response 1: The models constructed in the 1st level-I AEFR explained similar amounts of macroinvertebrate richness and had the maximum ability to explain how macroinvertebrate richness distributed (denoted “explaining ability”; geophysical landscape: RGL2 ≈ 60%, land use and site habitat: RLU2 and RSH2 ≈ 50%, and combined: RCB2 ≈ 75%).

Point-2: In Line 35-36, the original sentence “These results could be cost-effective tools to distinguish and realize the drivers of sensitive aquatic organisms at regional scales.“ revised as below.

Response 2: These results may provide cost-effective tools to distinguish and determine the drivers of sensitive aquatic organisms at regional scales.

Point-3: In Line 45-47, the original sentence “Nevertheless, freshwater ecosystems are the most threatened and some researchers indicate that the species extinction speed is five times higher than every other community.“ revised as below.

Response 3: Nevertheless, the biodiversity of freshwater ecosystems are the most threatened and some researchers indicate that the species extinction speed is five times higher than every other community.

Point-4: In Line 50-53, the original sentence “For the mobility, morphology, physiology, taxa sensitivity to disturbance, and the continuum response to environmental variables, benthic macroinvertebrates are often used for biological assessments of changes in environmental conditions in freshwater ecosystems via the single and multiple index and predicted model.“ revised as below.

Response 4: Benthic macroinvertebrates are often used for the biological assessment of changes in response to environmental conditions—such as mobility, morphology, physiology, taxa sensitivity to disturbance, and the continuum response to environmental variables—in freshwater ecosystems via single and multiple index and predicted models.

Point-5: In Line 59-61, the original sentence “Land use, from undisturbed to human-dominated landscapes, represent anthropogenic disturbance at the catchment scale and has been the research hotspot for the decades to illustrate the direct and indirect effects on freshwater ecosystems.“ revised as below.

Response 5: Land use, from undisturbed to human-dominated landscapes, represents anthropogenic disturbance at the catchment scale and has been used for decades to illustrate the direct and indirect anthropogenic effects on freshwater ecosystems.

Point-6: In Line 62-65, the original sentence “Previous studies have documented that land use can affect the function and processes of freshwater ecosystems through various pathways, including hydrological alternation, riparian clearing, loss of woody debris, input of excess sedimentation, and nutrient and containment, all of which affect the quality and availability of site habitats for aquatic organisms .“ revised as below.

Response 6: Previous studies have documented that land use can affect the function and processes of freshwater ecosystems through various pathways, including hydrological alteration, riparian clearing, loss of woody debris, input of excess sedimentation, and nutrient and containment, all of which affect the quality and availability of site habitats for aquatic organisms.

Point-7: In Line 65-67, the original sentence “Due to the greater direct effects of site-scale variables, the physical and chemical variables have a key impact on aquatic assemblages.“ revised as below.

Response 7: Owing to the greater direct effects of site-scale variables, physical and chemical variables have a key impact on aquatic assemblages.

Point-8: In Line 74-75, the original sentence “The differences between geophysical variables, land use, and human impacts among different ecoregions has been ignored.“ revised as below.

Response 8: The differences among geophysical variables, land use, and human impacts among different ecoregions have not been examined in previous research.

Point-9: In Line 83-87, the original sentence “This watershed has become a priority hotspot for biodiversity conservation in recent years due to the extinction of large endemic fish species (near half the historical species number now exists) and loss of biodiversity associated with the degraded water and physical habitat quality due to the intense deforestation and transformation from natural to agriculture and urban land use.“ revised as below.

Response 9: This watershed has become a hotspot for biodiversity conservation efforts in recent years, owing to the extinction of large endemic fish species (approximately half of the historically present species currently exist) and loss of biodiversity associated with degraded water quality and physical habitat quality as a result of intense deforestation and a transformation from undeveloped to agricultural and urban land use.

Point-10: In Line 97-99, the original sentence “The Liaohe River Basin (40°30′ ~ 45°10′ N, 117°00′ ~ 125°30′ E) is composed of two independent separated hydrographic nets, the Da-Liao and Hun-Tai River systems, which all afflux into the Bohai Sea.“ revised as below.

Response 10: The Liaohe River Basin (40°30′–45°10′ N, 117°00′–125°30′ E) is composed of two independent separate hydrographic nets, the Da-Liao and Hun-Tai River systems, which flow into the Bohai Sea.

Point-11: In Line 97-99, the original sentence “The maximal drop in altitude is more than 1200 m, decreasing from piedmont in a west and southeast direction to the central plain with the name of Liaohe River Plain, where the population of Liaoning Province is distributed.“ revised as below.

Response 11: The maximal drop in altitude is more than 1200 m, decreasing from piedmont in a west and southeast direction to the central plain with the name of Liaohe River Plain, where the large and medium-size cities and major grain-producing areas in Liaoning Province are distributed.

Point-12: In Line 114-115, the original sentence “The main stems and tributaries located in the 2nd level-I AEFR were almost in draught for years, so this AEFR was not included in the analysis..“ revised as below.

Response 12: for this sentence has already provided this information once above in Line 104-105, so this sentence is deleted.

Point-13: In Line 122-124, the original sentence “All the rivers were divided into segmentations based on the setting of intersection as the break point, which the intersection was gained by visualization.“ revised as below.

Response 13: All rivers were divided into segments on the basis of the setting of an intersection as the break point, and the intersection was determined by visualization.

Point-14: In Line 140-143, the original sentence “The final interpreted results were formed by amendment of the preliminary scheme within the combination of field investigation and geomorphology characteristic through a man-machine interaction method between artificial interpretation and closest classification method.“ revised as below.

Response 14: The final interpreted results were determined by amending the preliminary scheme on the basis of a combination of field investigation and geomorphology characteristics by using a man-machine interaction method between artificial interpretation and the closest classification method.

Point-15: In Line 183-187, the original sentence “For example, Longitude was highly positive correlated with distance from mouth (r = 0.916) when constructed the model for the 4th level-I AEFR, the two variables were all describe the spatial relative location and we consider the latter one contains more comprehensive meaning and it was always be an important environmental indicator in other relevant model constructions.“ revised as below.

Response 15: For example, Longitude was highly positive correlated with DisfrMouth (r = 0.916) in model construction for the 4th level-I AEFR. The two variables were all describe the spatial relative location and we consider DisfrMouth contains more comprehensive meaning, not only the physical location, but also the relative habitat quality for the sampling site. It was always be an important environmental driver in other relevant researches.

Point-16: In Line 194-196, the original sentence “Fourth, a combined model for all level environmental variables and partial linear regression were established and conducted to evaluate the relative importance of each environmental level on the richness.“ revised as below.

Response 16: Fourth, a combined model for all level environmental variables using partial linear regression was established and used to evaluate the relative importance of each environmental level on the richness.

Point-17: In Line 205-207, the original sentence “Although constructed and cropland use were not the dominant land use type, due to the serious soil erosion in Inner Mongolia, the substrate was characterized by a large sand proportion with the analogue situation in the 3rd level-I AEFR, with the maximum land use being cropland there.“ revised as below.

Response 17: Although constructed and cropland use were not dominant land use types, owing to the severe soil erosion in Inner Mongolia, the substrate was characterized by a large proportion of sand, analogously to the characterization in the 3rd level-I AEFR, where cropland accounted for the greatest land use.

Point-18: In Line 232, the original sentence “The explaining abilities of models in this AEFR were general poorer than the other two.“ revised as below.

Response 18: The explaining abilities of models in the 3rd AEFR were the lowest among all the AEFRs.

Point-19: In Line 234-235, the original sentence “The model in site habitat (25%) explained the most variance and geophysical landscape (11%) explained the least, while land use (14%) explained a moderate amount.“ revised as below.

Response 19: The site habitat model (25%) explained the most variance, and the geophysical landscape model (11%) explained the least, whereas the land use model (14%) explained a moderate amount.

Point-20: In Line 245-247, the original sentence “Six variables entered the site habitat model, including WaterTem and Boulder, which were positive, and the rest (Sand, Velocity, TN, and TP), which were all negative with richness.“ revised as below.

Response 20: Six variables were used in the site habitat model: WaterTem and boulder, which were positively associated with richness, and sand, velocity, TN, and TP, which were negatively associated with richness.

Point-21: In Line 282-284, the original sentence “Despite some weaker explaining ability of MLRs, our results showed that the key environmental variables that affect the macroinvertebrate assemblages differed a lot in level-I AEFRs of the Liaohe River Basin and were comparable with other studies.“ revised as below.

Response 21: Despite the relatively weak explaining ability of some MLRs, our results showed that the key environmental variables that affect macroinvertebrate assemblages substantially differed in level-I AEFRs of the Liaohe River Basin and were comparable to findings from other studies.

Point-22: In Line 296-298, the original sentence “DisfrMouth (positive) and UpcatchArea (negative) with richness explained the similar affection that the closer to the source, the better the retained site habitat quality and more biodiversity could be maintained.“ revised as below.

Response 22: DisfrMouth (positive) and UpcatchArea (negative) with richness explained the similar effect in which the closer to the source, the better the retained site habitat quality and the more biodiversity maintained.

Point-23: In Line 313-315, the original sentence “Another reason may be the longer the reach, the more area it flows through and as the accommodating ability of the containment increased, water quality decreased as the ultimate result.“ revised as below.

Response 23: Another reason for this effect may be that longer reaches flow through a grater area, and as the accommodating ability of the containment increases, water quality decreases.

Point-24: In Line 316-317, the original sentence “Different level-I AEFRs had different explaining abilities relating to land use and macroinvertebrate richness, with the strongest occurring in the 1st level-I AEFR.“ revised as below.

Response 24: Different level-I AEFRs had different explaining abilities regarding land use and macroinvertebrate richness; the strongest explaining ability was found in the 1st level-I AEFR.

Point-25: In Line 327-330, the original sentence “In the three MLR models, the variables which represent the microhabitat, substrate structure, and the conventional physicochemical variables all included into the explainer list. Substrate structure, especially the composition of diverse cobblestone, has been proven to be the key factor influencing macroinvertebrate structure.“ revised as below.

Response 25: In the three MLR models, the variables representing the microhabitat, substrate structure, and conventional physicochemical variables were all included in the explainer list. Substrate structure, especially the composition of diverse cobblestone, has been shown to be the key factor influencing macroinvertebrate structure.

Point-26: In Line 368-370, the original sentence “The relationship between land use and richness was not, or only feebly, detected, the possibility may be of the covariance effects or the lower variability in Hun-Tai up-streams.“ revised as below.

Response 26: The relationship between land use and richness was not detected or only weakly detected, possibly because of covariance effects or the lower variability in Hun-Tai up-streams.

Reviewer 2 Report

Dear authors, I think that the manuscript can be published after minor revision reported in the sended version.

Author Response

Response to Reviewer 2 Comments

Point-1: The article topic has been revised according to the comment as below.

Response 1: The relative importance of spatial scale variables for macroinvertebrate richness in different aquatic ecological function regions in the Liaohe River Basin, a large temperate river basin in northeast China.

Point-2: The amount of collected macroinvertebrate samples has been added in Line 158-159, and revised as below.

Response 2: A total of 1628 macroinvertebrate samples, one qualitative and three quantitative samples for each site, were collected.

Point-3: Line space of all tables have been revised according to the comment.

Response 3: Line space of all tables have been revised as single space. 

Point-4: Characters in Table 4 need to be larger according to the comment.

Response 4: The authors tried to change the characters more larger, however the table format changed simultaneously. So, in order to keep the Table 4 in one page, the authors suggest to retain the current format of Table 4.
